# Study on a Three-Step Rapid Assembly of Zolpidem and Its Fluorinated Analogues Employing Microwave-Assisted Chemistry

**DOI:** 10.3390/molecules25143161

**Published:** 2020-07-10

**Authors:** Nikola Fajkis, Monika Marcinkowska, Beata Gryzło, Anna Krupa, Marcin Kolaczkowski

**Affiliations:** Faculty of Pharmacy, Jagiellonian University Medical College, 9 Medyczna Street, 30-688 Kraków, Poland; nikola.fajkis@doctoral.uj.edu.pl (N.F.); beata.gryzlo@uj.edu.pl (B.G.); a.krupa@uj.edu.pl (A.K.); marcin.kolaczkowski@uj.edu.pl (M.K.)

**Keywords:** zolpidem, GABA-A receptor, imidazo[1,2-*a*]pyridine, microwave supported synthesis

## Abstract

We developed an efficient microwave-assisted three-step synthesis of zolpidem and its fluorinated analogues **1**–**3**. The procedure relays on the utilization of easily accessible and inexpensive starting materials. Our protocol shows superior performance in terms of yield and purity of products, compared to conventional heating systems. Notably, the total time needed for reaction accomplishment is significantly lower comparing to oil bath heating systems. Finally, we have performed a detailed study on the preparation of zolpidem tartrate salt **I**, and we assessed its particle-sizes using a polarizing microscope. Our goal was to select the appropriate method that generates the acceptable particle-size, since the solid-size directly influences solubility in biological fluids and further bioavailability. We believe that the disclosed procedure will help to produce a lab-scale quantity of zolpidem and its fluorinated derivatives **1**–**3**, as well as zolpidem tartrate salt **I,** with suitable fine-particle size for further biological experimentation.

## 1. Introduction

Zolpidem **1** is one of the most widely prescribed hypnotic medications in the United States, with an annual prescription rate of nearly 39 million [1,2]. It was developed in the late 1980s [3] and 1990s as a subtle alternative to benzodiazepines, which could rapidly promote sleep, while retaining the negligible risk of impaired daily functioning [4]. Zolpidem binds selectively to GABA-A (γ-aminobutyric acid) receptors containing the α1 subunit, and further mediates synaptic GABA-ergic signaling [5]. During the past decade, zolpidem has featured in several curious case reports that documented the reversal of brain stroke symptoms [6], improvement of motor dysfunction in Parkinson’s disease patients [7] or the effective reduction of chronic pain [8]. All these unique features are attributed to the enhanced synaptic GABA-ergic currents in specific brain regions. In addition, our group found that zolpidem **1** [9], as well as its fluorinated analogues **2**, **3** [10,11] (Figure 1), display prominent antipsychotic activity in animal models of schizophrenia. These findings opened new prospects for the therapeutic utility of selective α1-GABA-A receptor ligands in the treatment of various pathologic conditions, besides insomnia. In this regard, a plethora of experiments have been recently conducted by many research groups, using zolpidem as a pharmacological tool to probe the activity of α1-GABA-A receptor. As zolpidem is a short-acting ligand characterized by fast elimination from the organism, this makes it merely appropriate for acute or short-term biological experiments. The fluorinated analogues of zolpidem, developed in our lab, are characterized by promising metabolic stability and a longer duration of biological activity [10,11], thus, are relevant tools for the purpose of chronic treatment studies. Consequently, zolpidem **1**, and its fluorinated analogues **2**, **3**, are highly desirable pharmacological tools for biological investigations, and the development of synthetic methods that would give fast and easy access to **1**–**3** are highly warranted.

Recent advances in heterocyclic chemistry revealed that it is possible to obtain zolpidem and possibly its fluorinated analogues by applying diverse one-pot protocols or intramolecular double aza-Michael addition of 2-aminopyridines to Morita-Baylis-Hillman (MBH) acetates [12]. However, these methods require usage of special technological devices, employ hazardous reagents and necessitate fluorinated starting materials, which are not easily available from commercial sources, and thereby require extensive preparation [13]. Alternatively, zolpidem and its analogues might be prepared by lengthy six to eight step procedures, which generate large amounts of waste, require expensive starting materials, give moderate yields and are narrowed in scope [14].

Recently we reported a practical three-step synthesis of **2**–**3**, based on the incorporation of N,N-dimethylaminoacetamide backbone into position 3 of the 2-phenylimidazo[1,2-a]-pyridine framework [10] (Scheme 1). The main advantage of this pathway is the direct use of α-bromoacetophenones and fluoropyridine-2-amines, which represent relatively cheap and the most facilely available starting materials in a broad range of fluorination degrees. The method proved to be viable, but unexpectedly moderately yielding, somewhat lengthy and providing products with high amount of trace impurities. Mounting evidence shows that the application of microwave-assisted techniques significantly accelerate reaction rates, provide higher yields and product purities, as well as lowering the processing cost. Furthermore, when it applies minimal solvent amounts, it helps to achieve cleaner reactions [15,16,17]. Thus, we reasoned that a slight modification of our three-step protocol and employment of microwave irradiation would be particularly advantageous, in terms of accelerated reaction time and higher yields and purities of final products.

Herein, we report a fast, high yielding, experimentally simple and less cumulative energy demanding (CED) synthesis of zolpidem **1** and its fluorinated analogues **2**, **3**. The developed protocol portrayed in Scheme 1 takes advantage of microwave-enhanced chemistry, is operatively simple, reactions take place smoothly and effectively. Our microwave-assisted synthesis shows superior performance compared to the reactions and yields achieved with conventional heating. An additional challenging issue was the preparation of zolpidem salt with tartaric acid, which accounts for suitable solubility in biological fluids and provides adequate bioavailability [18]. Therefore, we attempted to develop an effective procedure for achieving zolpidem tartrate salt and performed a detailed characterization of the achieved product using a polarizing microscope.

## 2. Results and Discussion

### 2.1. Synthesis

We began the optimization process from the preparation of 2-phenylimidazo[1,2-*a*]pyridine cores **4**–**6** (step 1, Table 1) via the condensation of suitably substituted 2-aminopyridines and α-bromoacetophenones, applying microwave-enhanced chemistry. Despite the apparent simplicity of the process, the initial results were not very satisfactory, as we found that the reactions were highly sensitive to the type of solvent used. Therefore, we decided to study the process and screen several different microwave absorbing solvents (Table 1). Considering an economic standpoint, we selected 5-methylpyridin-2-amine and 2-bromo-1-(p-tolyl)ethan-1-one as the model reaction for optimization. We found that with the use of high microwave absorbing, polar solvents such as methanol did not induce conversion to the desired products (Table 1). Quite surprisingly, the use of nonpolar solvent, such as toluene, exhibited a remarkable enhancement of the reactivity. Clearly the microwave irradiation accelerated the condensation process in the presence of low microwave absorbing solvent such as toluene. We suspect that the improved performance in toluene mainly depended on the formation of ‘hot spots’, produced by increased microwave absorption at the surface of the insoluble NaHCO_3_ particles [16,19,20].

Next, using the optimal conditions set in the microwave apparatus, we performed the condensation among fluorinated 2-aminopyridines and α-bromoacetophenones, and found that fluorinated starting materials were also amenable substrates for this transformation with good product yields (71–91%, Table 2). As the multicomponent reactions performed under microwave heating are highly dependent on the concentration [16] of the reaction mixtures, as an ultimate factor, we decided to compare the reaction yields heated in 0.1 M (0.3 mmol/3 mL) and 0.06 M (0.3 mmol/5 mL) of toluene. In this particular case, for all reactions, higher yields were observed in more condense settings; 0.1 M (0.3 mmol/3 mL). Overall, the optimal conditions found for **4**–**6** were: 1 equiv. pyridin-2-amine and α-bromoacetophenones, 1.5 eq of sodium bicarbonate heating at 100 °C in toluene (concentration: 0.1 M) for 40 min (detailed description provided in the experimental section). Notably, the synthesis of 2-phenylimidazo[1,2-*a*]pyridines in conventional oil-bath experiments, typically requires overnight heating under the reflux. Transferring the synthesis under microwave conditions not only accelerated the reaction time, but also provided higher product purities and remarkably higher isolated yields (Table 2).

The second step involves functionalization of 2-phenylimidazo[1,2-*a*]pyridine cores at the 3 position (Step 2, Table 3). This transformation has been described in the literature only occasionally, and, in some cases, limited experimental information was available. In general, α-hydroxyacetamide intermediates **7**–**9** can be obtained by reacting 2-phenylimidazo[1,2-*a*]pyridines with hemi-hydrate of *N*,*N*-dimethylglyoxylamide [21], acetal of *N*,*N*-dimethyglyoxamides [22] or *N*,*N*-dimethyl-2,2-dichloroacetamide (in a two-step reaction [23]). However, these reagents are unstable, require in situ preparation in a very laborious procedures and give the desired α-hydroxyacetamide intermediates in very low yields. Previously, we reported that a stable form of N,N-dimethylglyoxylamide might be achieved by direct oxidation of *N*,*N*,*N*′,*N*′-tetramethyl-L-tartramide [10]. Next, *N*,*N*-dimethylglyoxylamide can be reacted with suitable 2-phenylimidazo[1,2-*a*]pyridines to form α-hydroxyacetamide intermediates **7**–**9**, simply by heating starting materials in the presence of acetic acid. The procedure proved to be advantageous upon traditional heating system, however the α-hydroxyacetamide intermediates **7**–**9** were isolated after 12 h with 50–62% yield, together with relevant amounts of acetate by-products. When we applied equal conditions in microwave reactors, the product could be obtained easily after 1h, however, the acetate by-products were also forming virtually instantaneously. To overcome these difficulties, we sought to perform the reaction, reducing the amount of acetic acid to the minimum amount required, to catalyze the process and heat the reaction in MW. Gratifyingly, employing these settings we were able to suppress almost entirely the formation of ester by-products, and to obtain α-hydroxyacetamide intermediates **7**–**9** within 1h, as the major products in very good yields 81–88%. Moreover, we found that the desired products could be isolated easily from the reaction mixture by filtrating the precipitated product (details provided in the experimental section).

With the α-hydroxyacetamide precursors **7**–**9** in hand, we proceeded with the synthesis of the final products **1**–**3**. The final dehydroxylation was performed in MW, in the presence of phosphorus tribromide added in an excess to the reaction mixture (Table 4). During the reaction a precipitable salt is formed which can be collected by filtration and further hydrolyzed during the workup, to give the final products **1**–**3**. This process has been described previously to occur using various solvents (polar or halogenated solvents) in the traditional heating systems. We decided to transfer the experimental setting and perform the reaction under MW heating, starting with the screening of a variety of solvents. We observed that, when the reaction was carried out in dichloroethane (at 120 °C), the substrates reacted vigorously, forming high amounts of trace impurities (according to HPLC and NMR), which are difficult to identify and inseparable from the main product. Moreover, after several purification attempts the final products still appeared as intense yellow solids, which is not with agreement with the previous literature reports, which describe it as white solids. A possible explanation of the low performance of PBr_3_ in dichloroethane is related with the fact that dichloroethane tend to decompose under a high temperature, forming a mixture of various products [24,25], which further react with the remaining components of the reaction mixture, leaving a high amount of impurities. To circumvent this undesired event, we opted to perform the reaction, replacing dichloroethane with non-halogenated solvents. Satisfactorily, we found that the use of polar solvents such as tetrahydrofuran (THF) or dioxane in MW proved to be extremely rewarding, which resulted in strikingly clean reactions and a high purity of the final products **1**–**3**. The substrates reacted smoothly affording the final products in very clean reactions with good yields and high purity. We observed that performing the reaction in the presence of THF, heating for 45 min at 50 °C in MW proved to be slightly more effective over reactions heated at 110 °C in dioxane (Table 4). These differences were particularly noticeable for 2-phenylimidazo[1,2-*a*]pyridines bearing 6-methyl **1** and 4′-F **3**.

We attempted also to transform the intermediate alcohols to final acetamides exploring other conditions. Several different dehydroxylation procedures were tested, including I_2_/PPh_3_ [26], Pd/HCOOH [27], TMSCl/NaI [2]. However, upon all investigated conditions, dehydroxylation took place to a very small extent and the desired acetamides were always obtained as minor products in a mixture with the carboxylic acids delivered by hydrolysis of **7**–**9**, along with other unidentified side-products. Nevertheless, the dehydroxylation applying phosphorus tribromide in THF, in our hands, proved to be the most fruitful. The final products can be obtained within 45 min, in very clean reactions with good yields, while potentially hazardous unreacted materials can be easily removed during the filtration step and work up.

Altogether, performing the three-step synthesis of **1**–**3** under microwave heating resulted in clean reactions, increased yields, significantly reduced the amount of solvents and effectively shortened the time needed for transformation and reduced concurrent side reactions.

### 2.2. Preparation of Water-Soluble Salts

With the lab-scale quantities of zolpidem **1**, we next proceeded with the preparation of corresponding organic salts. We chose zolpidem as a model compound for investigating the formation of suitable salt, because it has been recently widely used by researchers as a pharmacological tool for biological investigation. Zolpidem free base **1** has been reported to be almost insoluble in water (~0.1mg/mL) [28,29], which, in turn, may result in poor bioavailability and inconsistent in vivo responses [30]. This inconvenient feature has prompted the pharmaceutical industry towards the composition of suitable salts characterized by improved solubility [18,31]. In fact, zolpidem appears in various pharmaceutical compositions as a salt with natural L(+)tartaric acid, where the molar ratio of zolpidem and tartaric acid in the salt is 2:1 [28,32] (Table 5, **I**). Zolpidem tartrate **I** possesses advantageous solubility in water (18mg/mL) and other similar pharmaceutically acceptable solvents. Based on the literature reports, we proceeded with the preparation of zolpidem tartrate (2:1) by mixing both substrates in methanol and allowing to settle the crystals overnight (Method A). Direct filtration of the resulting crystals afforded the desired salt (**I**) with high purity, however, with a considerably low yield. In the second attempt, we straightforwardly evaporated the reaction mixture (Method B). However, we could confidently predict that such a procedure resulted in a notably higher yield; also, the salt contained higher amounts of various impurities, inseparable by further crystallization attempts. After this explorative attempt, we changed the strategy and decided to accelerate the formation of crystals by subsequent addition of a precipitant (diethyl ether), which had been previously employed in our labs to give excellent results [10,33,34]. The addition of Et_2_O proved to be particularly rewarding, as the salt was separated in a good yield and excellent purity (Table 5).

### 2.3. Evaluation of the Particles’ Size

An additional aspect that may equally influence compound’s solubility and bioavailability is the particle’s size, shape and texture. These features are crucial factors, from the technological point of view, while thinking of developing first dosage forms for preclinical studies [35]. In general, fine non-agglomerated particles have a higher surface area, and therefore a higher solubility and dissolution rate than coarse particles or aggregates. As a result, many slowly dissolving, poorly soluble drugs are commercially available in the micronized form [35]. Yet, the micronization of hydrophobic particles may also cause their aggregation, which finally makes solubility enhancement impossible. Apart from solubility concerns, the crystal habit determines also other important features of compounds, such as powder flow, or sedimentation and caking in suspensions, which should be taken into account upon a new dosage form development [18,31]. During the experimentation process, we noticed that the crystals of formed salt (I) can be altered, depending on the applied conditions. While a direct crystallization afforded thick compact agglomerates formed of primary lath particles (Method A), an alternative method based on addition of Et_2_O (Method C) resulted in irregular fine primary particles, which formed less coherent and much smaller aggregates. It was assumed that the selection of the most favorable morphology of crystals could be crucial to prepare formulations for biological studies. We reasoned that, otherwise, one would risk to acquire apparently negative results associated with poor solubility and bioavailability, rather than the real lack of biological activity.

Thus, the crystal habit of developed particles was analyzed in a more detailed way, using polarized light microscopy. The images of the particles obtained by Method A and C are shown in Figure 2. The product developed in Method B based on a direct evaporation of the reaction mixture was omitted due to the large number of impurities.

As can be seen from Figure 2, all examined products showed characteristic birefringence testifying their crystallinity. However, the crystals varied significantly in both size and shape. Overnight crystallization from methanol (Method A) resulted in agglomerated lath particles of about 10–20 μm in length (Figure 2A,B). In contrast, the addition of the precipitant (Et_2_O) delivered mostly non-agglomerated fine particles of less than 5 μm in length (Figure 2C,D). Since the tendency to form agglomerates and the particle size may have a fundamental impact on the properties of biologically active molecules, controlling their solubility and bioavailability, we believe that Method B can be a promising approach to deliver a high quality zolpidem tartrate, which may be used as a pharmacological probe for various biological studies.

## 3. Materials and Methods

### 3.1. General Methods

Unless otherwise indicated, all the starting materials were obtained from commercial suppliers (Merck, Alfa Aesar or Fluorochem) and were used without further purification. Analytical thin-layer chromatography (TLC) was performed on Merck Kieselgel 60 F_254_ (0.25 mm) pre-coated aluminum sheets (Merck, Darmstadt, Germany). Visualization was performed with a 254 nm UV lamp. Column chromatography was performed using silica gel (particle size 0.063–0.200 mm; 70–230 Mesh ATM) purchased from Merck. The UPLC-MS or UPLC-MS/MS analyzes were run on UPLC-MS/MS system comprising Waters ACQUITY^®^ UPLC^®^ (Waters Corporation, Milford, MA, USA), coupled with Waters TQD mass spectrometer (electrospray ionization mode ESI with tandem quadrupole). Chromatographic separations were carried out using the ACQUITY UPLC BEH (bridged ethyl hybrid) C_18_ column: 2.1 × 100 mm and 1.7 µm particle size. The column was maintained at 40 °C and eluted under gradient conditions using 95% to 0% of eluent A over 10 min, at a flow rate of 0.3 mL/min. Eluent A: water/formic acid (0.1%, v/v); eluent B: acetonitrile/formic acid (0.1%, v/v). A total of 10 µl of each sample were injected, and chromatograms were recorded using a Waters eλ PDA detector. The spectra were analyzed in the range of 200–700 nm, with a 1.2 nm resolution, and at a sampling rate of 20 points/s. MS detection settings of Waters TQD mass spectrometer were as follows: source temperature 150 °C, desolvation temperature 350 °C, desolvation gas flow rate 600 L/h, cone gas flow 100 L/h, capillary potential 3.00 kV and cone potential 20 V. Nitrogen was used for both nebulizing and drying. The data were obtained in a scan mode ranging from 50 to 1000 *m*/*z* at 0.5 s intervals; 8 scans were summed up to obtain the final spectrum. Collision activated dissociation (CAD) analyses were carried out with the energy of 20 eV, and all the fragmentations were observed in the source. Consequently, the ion spectra were obtained in the range from 50 to 500 *m*/*z*. MassLynx V 4.1 software (Waters) was used for data acquisition. Standard solutions (1 mg/mL) of each compound were prepared in a mixture comprising analytical grade acetonitrile/water (1/1, v/v). The UPLC/MS purity of all the test compounds and key intermediates was determined to be >95%. ^1^H NMR, ^13^C NMR, and^19^F NMR spectra were obtained in a Joel spectrometer (Jeol, 500 MHz), in CDCl_3,_ CD_3_OD or DMSO operating at 500 MHz (^1^H NMR), 126 MHz (^13^C NMR), and 471 MHz (^19^F NMR). The chemical shifts are reported in ppm and were referenced to the residual solvent signals (CHLOROFORM-d ^1^H: 7.26 ppm, ^13^C: 77.16 ppm; CD_3_OD ^1^H: 3.31 ppm, ^13^C: 49.00 ppm), coupling constants are reported in hertz (Hz). The *J* values are expressed in Hertz. Signal multiplicities are represented by the following abbreviations: s (singlet), br.s (broad singlet), d (doublet), dd (doublet of doublets), dt (doublet of triplets), dtd (doublet of triplet of doublets), t (triplet), td (triplet of doublets), tdd (triplet of doublet of doublets), q (quartet), quin (quintet), m (multiplet). This is in agreement with the structural data reported in the literature [10]. The spectral data of final compounds were previously reported by our group [10], however, they were recorded on Varian 300 MHz apparatus, therefore, we decided to record them again using Jeol, 500 MHz. The microwave-assisted reactions were performed using a CEM Discover System 908010 microwave apparatus (Matthews, NC, USA), operating at a frequency of 2.45 GHz with continuous irradiation power from 0 to 200 W. Tetrahydrofuran (THF) was distilled under nitrogen immediately before use. The drying agent used for THF was sodium/benzophenone ketyl. Dioxane dry, toluene dry and methanol dry were purchased from commercial sources (Sigma Aldrich, Poznan, Poland). Images of zolpidem tartrate I were taken using an RT series camera (Opta-Tech, Warsaw, Poland) and the polarized light microscope Hund Wetzlar type H600/12 (Helmut Hund GmbH, Wetzlar, Germany).

### 3.2. Compound Characterization

*6-methyl-2-(p-tolyl)imidazo[1,2-a]pyridine (4).* To a solution of 5-methylpyridin-2-amine (314 mg, 3 mmol, 1 eq) in toluene (3 mL), 4-methylbenzoyl bromide (636 mg, 3 mmol, 1 eq) and sodium hydrogen carbonate (378 mg, 4.5 mmol, 1.5 eq) was added. The mixture was irradiated at 110 °C for 40 min in CEM-Discover system, operating at a frequency of 2.445 GHz, with continuous irradiation power from 0 to 200 W. After that time, the reaction mixture was extracted with EtOAc, dried over Na_2_SO_4_ and concentrated under reduced pressure. Purification by column chromatography using (Hex:EtOAc 7:3) as eluent, gave **4** (585 mg, 90%) as a light yellow powder mp 200.8–203.7 °C, Rf = 0.31 (Hex:EtOAc 7:3); ^1^H NMR (500 MHz, CDCl_3_) δ = 7.89 (s, 1 H), 7.85–7.82 (m, 2 H), 7.74 (s, 1 H), 7.54 (d, J = 9.2 Hz, 1 H), 7.24 (d, J = 8.0 Hz, 2 H), 7.01 (dd, J = 1.7, 9.2 Hz, 1 H), 2.39 (s, 3 H), 2.31 (d, J = 1.1 Hz, 3 H); ^13^C NMR (126MHz, CDCl_3_) δ = 145.6, 144.7, 137.7, 131.0, 129.5, 127.9, 125.9, 123.4, 122.1, 116.8, 107.6, 21.4, 18.2; EI MS *m/z:* 223.0 [M + H^+^]. Alternatively, the reaction performed using 5 mL of toluene afforded the product **4** with 71% yield.

*2-(4-fluorophenyl)—6-methylimidazo[1,2-a]pyridine (5).* To a solution of 5-methylpyridin-2-amine (314 mg, 3 mmol, 1 eq) in toluene (3 mL), 4-fluorobenzoyl bromide (648 mg, 3 mmol, 1 eq) and sodium hydrogen carbonate (378 mg, 4.5 mmol, 1.5 eq) was added. The mixture was irradiated at 110 °C for 40 min in the CEM-Discover system, operating at a frequency of 2.445 GHz, with continuous irradiation power from 0 to 200 W. The reactor was cooled down to room temperature and depressurized. The reaction mixture was extracted with EtOAc, dried over Na_2_SO_4_ and concentrated under reduced pressure. Purification by column chromatography using (Hex:EtOAc 7:3) as eluent gave **5** (560 mg, 82%) as a light yellow powder; mp 198.9–202.5 °C; Rf = 0.31 (Hex:EtOAc 7:3); ^1^H NMR (500 MHz, CHLOROFORM-d) δ = 7.88 (dd, J = 5.4, 9.0 Hz, 3 H), 7.69 (s, 1 H), 7.50 (d, J = 9.2 Hz, 1 H), 7.09 (t, J = 8.8 Hz, 2 H), 7.00 (dd, J = 1.6, 9.2 Hz, 1 H), 2.30 (s, 3 H); ^13^C NMR (126MHz, CHLOROFORM-d) δ = 163.4 (d, *J* = 248.7 Hz), 144.9 (d, *J* = 24.0 Hz), 130.3, 128.0, 127.7 (d, *J* = 3.8 Hz), 123.4, 122.2, 116.9, 115.8 (d, *J* = 3.1 Hz), 107.6, 18.2; ^19^F NMR (471MHz, CHLOROFORM-d) δ = −114.4 (s, 1 F); EI MS *m*/*z:* 227.0 [M + H^+^]; Alternatively, the reaction performed using 5 mL of toluene afforded the product **5** with 75% yield.

*6-fluoro-2-(4-fluorophenyl)imidazo[1,2-a]pyridine (6).* To a solution of 5-fluoropyridin-2-amine (336 mg, 3 mmol, 1 eq) in toluene (3 mL), 4-fluorobenzoyl bromide (648 mg, 3 mmol, 1 eq) and sodium hydrogen carbonate (378 mg, 4.5 mmol, 1.5 eq) was added. The mixture was irradiated at 110 °C for 40 min in the CEM-Discover system, operating at a frequency of 2.445 GHz, with continuous irradiation power from 0 to 200 W. The reactor was cooled down to room temperature and depressurized. The reaction mixture was extracted with EtOAc, dried over Na_2_SO_4_ and concentrated under reduced pressure. Purification by column chromatography using (Hex:EtOAc 7:3) as eluent gave **6** (616 mg, 89%) as a brown powder; mp 171.2–175.3 °C; Rf = 0.45 (Hex:EtOAc 7:3); EI MS *m*/*z:* 231.0 [M + H^+^]; ^1^H NMR (500 MHz, CHLOROFORM-d) δ = 8.07–8.00 (m, 1 H), 7.93–7.83 (m, 2 H), 7.79 (s, 1 H), 7.64–7.53 (m, 1 H), 7.15–7.05 (m, 3 H); ^13^C NMR (126MHz, CHLOROFORM-d) δ = 163.0 (d, *J* = 245.9 Hz), 153.3 (d, *J* = 236.4 Hz), 146.3, 143.5, 129.7 (d, *J* = 3.5 Hz), 127.8 (d, *J* = 7.1 Hz), 117.9 (d, *J* = 9.6 Hz), 116.9 (d, *J* = 25.3 Hz), 115.8 (d, *J* = 21.7 Hz), 112.2 (d, *J* = 40.1 Hz), 109.2; ^19^F NMR (471MHz, CHLOROFORM-d) δ = −113.6 (s, 1 F), −140.32 (s, 1 F); EI MS *m*/*z:* 231.0 [M + H^+^]. Alternatively, the reaction performed using 5 mL of toluene afforded the product **6** with 86% yield.

*2-hydroxy-N,N-dimethyl-2-(6-methyl-2-(p-totyl)imidazo[1,2-α]pyridin-3-yl)acetamide (7).* To a solution of 6-methyl-2-(p-tolyl)imidazo[1,2-α]pyridine **4** (672 mg, 3 mmol, 1 eq) in toluene (4 mL) and acetic acid (1 mL), *N*,*N*-dimethyl-2-oxoacetamide (756 mg, 7.5 mmol, 2.5 eq) was added. The mixture was irradiated at 65 °C for 60 min in the CEM-Discover system, operating at a frequency of 2.445 GHz, with continuous irradiation power from 0 to 200 W. The reactor was cooled down to room temperature and depressurized. The reaction mixture was extracted with EtOAc, dried over Na_2_SO_4_ and concentrated under reduced pressure. The product was separated from remaining mixture by filtration and the remaining residue was purified by column chromatography using (Et_2_O:DCM: MeOH 2:7.5:0.5) as eluent to give **7** (790 mg, 81%) as a white powder; mp 170.2–172.3 °C; Rf = 0.73 (Et_2_O:DCM:MeOH 2:7.5:0.5); ^1^H NMR (500 MHz, CHLOROFORM-d) δ = 7.96–7.92 (m, 1 H), 7.66–7.63 (m, 2 H), 7.53 (d, *J* = 9.2 Hz, 1 H), 7.29–7.24 (m, 2 H), 7.06 (dd, *J* = 1.7, 9.2 Hz, 1 H), 5.69 (s, 1 H), 2.84 (s, 3 H), 2.38 (s, 3 H), 2.34–2.32 (m, 3 H), 2.30 (s, 3 H); ^13^C NMR (126 MHz, CHLOROFORM-d) δ =170.4, 146.0, 144.7, 138.3, 130.1, 129.6, 128.9, 128.4, 122.9, 122.6, 117.0, 115.7, 64.0, 36.8, 36.2, 18.5; EI MS *m*/*z:* 323.9 [M + H^+^].

*2-(2-(4-fluorophenyl)-6-methylimidazo[1,2-α]pyridin-3-yl)-2-hydroxy-N,N-dimethylacetamide (8).* To a solution of 6-fluoro-2-(4-fluorophenyl)imidazo[1,2-a]pyridine **5** (684 mg, 3 mmol, 1 eq) in toluene (4 mL) and acetic acid (1 mL), *N*,*N*-dimethyl-2-oxoacetamide (758 mg, 7.5 mmol, 2.5 eq) was added. The mixture was irradiated at 65 °C for 60 min in the CEM-Discover system, operating at a frequency of 2.445 GHz, with continuous irradiation power from 0 to 200 W. The reactor was cooled down to room temperature and depressurized. The reaction mixture was extracted with EtOAc, dried over Na_2_SO_4_ and concentrated under reduced pressure. The product was separated from remaining mixture by filtration and the remaining residue was purified by column chromatography using (Et_2_O: DCM:MeOH 2:7.5:0.5) as eluent to give **8** (851 mg, 86%) as a white powder; mp 183.5–185.2 °C); Rf = 0.67 (Et_2_O:DCM:MeOH 2:7.5:0.5); ^1^H NMR (500 MHz, CHLOROFORM-d) δ = 7.94 (d, *J* = 1.1 Hz, 1 H), 7.82–7.66 (m, 2 H), 7.52 (d, *J* = 9.2 Hz, 1 H), 7.18–7.12 (m, 2 H), 7.09 (dd, *J* = 1.7, 9.2 Hz, 1 H), 5.63 (s, 1 H), 2.84 (s, 3 H), 2.34 (s, 3 H), 2.31 (d, *J* = 1.1 Hz, 3 H); ^13^C NMR (126MHz, CHLOROFORM-d) δ = 170.1, 163.0 (d, *J* = 248.9 Hz), 144.9 (d, *J =* 24.9 Hz), 130.8, 130.0, 129.9 (d, *J* = 3.7 Hz), 128.7, 122.8, 117.0, 115.9 (d, *J* = 3.2 Hz), 115.7, 63.8, 36.8, 36.1, 29.4, 18.5; ^19^F NMR (471 MHz, CHLOROFORM-d) δ = −113.2 (s, 1 F); EI MS *m*/*z:* 328.0 [M + H^+^].

*2-6-fluoro-2-(4-fluorophenyl)imidazo[1,2-α]pyridin-3-yl)-2-hydroxy-N,N-dimethylacetamide (9).* To a solution of 2-(4-fluorophenyl)-6-methylimidazo[1,2-a]pyridine **6** (696 mg, 3 mmol, 1 eq) in toluene (4 mL) and acetic acid (1 mL), *N*,*N*-dimethyl-2-oxoacetamide (758 mg, 7.5 mmol, 2.5 eq) was added. The mixture was irradiated at 65 °C for 60 min in the CEM-Discover system, operating at a frequency of 2.445 GHz, with continuous irradiation power from 0 to 200 W. The reactor was cooled down to room temperature and depressurized. The reaction mixture was extracted with EtOAc, dried over Na_2_SO_4_ and concentrated under reduced pressure. The product was separated from remaining mixture by filtration and the remaining residue was purified by column chromatography using (Et_2_O: DCM:MeOH 2:7.5:0.5) as eluent to give **9** (921 mg, 88%) as a brown powder; mp 172.2–174.5 °C; Rf = 0.70 (Et_2_O:DCM:MeOH 2:7.5:0.5); ^1^H NMR (500 MHz, CHLOROFORM-d) δ = 8.20–8.13 (m, 1 H), 7.77–7.70 (m, 2 H), 7.61 (dd, *J* = 5.2, 9.7 Hz, 1 H), 7.22–7.12 (m, 3 H), 5.66 (s, 1 H), 2.89 (s, 3 H), 2.36 (s, 3 H); ^13^C NMR (126 MHz, CHLOROFORM-d) δ = 169.3, 163.3 (d, *J* = 248.8 Hz), 153.2 (d, *J* = 239.3 Hz), 146.2, 143.3, 130.7 (d, *J* = 7.6 Hz), 129.5 (d, *J* = 3.4 Hz), 118.1 (d, *J* = 8.4 Hz), 117.6 (d, *J* = 2.1 Hz), 116.1 (d, *J* = 21.7 Hz), 112.4, 112.1, 63.6, 36.8, 36.1; ^19^F NMR (471 MHz, CHLOROFORM-d) δ = −126.6 (s, 1 F), −138.2 (s, 1 F); EI MS *m*/*z:* 331.9 [M + H^+^].

*N,N*-dimethyl-2-(6-methyl-2-(*p*-totyl)imidazo[1,2-α]pyridin-3-yl)acetamide (1). To a solution of 2-hydroxy-*N*,*N*-dimethyl-2-(6-methyl-2-(p-totyl)imidazo[1,2-a]pyridin-3-yl)acetamide **7** (50 mg, 0.151 mmol, 1 eq) in THF (3 mL), phoshorus tribromide (0.039 mL, 0.393 mmol, 2.6 eq) was added. The mixture was irradiated at 90 °C for 45 min in the CEM-Discover system, operating at a frequency of 2.445 GHz, with continuous irradiation power from 0 to 200 W. The reactor was cooled down to room temperature and depressurized. Next, hexane, water and a saturated solution of sodium hydrogen carbonate was added. The reaction mixture was extracted with EtOAc, dried over Na_2_SO_4_ and concentrated under reduced pressure. Purification by column chromatography using (Et_2_O:DCM:MeOH 2:7.5:0.5) as eluent gave **1** (33 mg, 71%) as a light green powder; mp 184.2–186.8 °C, Rf = 0.48 (Et_2_O: DCM:MeOH 2:7.5:0.5); ^1^H NMR (500 MHz, CHLOROFORM-d) δ = 7.94 (d, *J* = 1.1 Hz, 1 H), 7.65 (d, *J* = 8.6 Hz, 2 H), 7.53 (d, *J* = 9.2 Hz, 1 H), 7.28–7.23 (m, 2 H), 7.06 (dd, *J* = 1.7, 9.2 Hz, 1 H), 4.09 (s, 2 H), 2.84 (s, 3 H), 2.38 (s, 3 H), 2.34–2.32 (s, 3 H), 2.30 (s, 3 H); ^13^C NMR (126 MHz, CHLOROFORM-d) δ = 170.4, 146.0, 144.7, 138.3, 130.9, 129.6, 128.9, 128.4, 122.9, 122.6, 117.0, 115.7, 36.8, 36.2, 30. 6, 21.4, 18.5, EI MS *m*/*z:* 308.0 [M + H^+^]. Alternatively, reaction performed in dioxane (3mL) gave 1 with 58% yield.

*2-(2-(4-fluorophenyl)-6-methylimidazo[1,2-α]pyridin-3-yl)-2-hydroxy-N,N-dimethylacetamide (2).* To a solution of 2-(2-(4-fluorophenyl)-6-methylimidazo[1,2-α]pyridin-3-yl)-2-hydroxy-*N*,*N*-dimethylacetamide **8** (50 mg, 0.153 mmol, 1 eq) in dioxane (3 mL), phoshorus tribromide (0.039 mL, 398 mmol, 2.6 eq) was added. The mixture was irradiated at 90 °C for 45 min in the CEM-Discover system, operating at a frequency of 2.445 GHz, with continuous irradiation power from 0 to 200 W. The reactor was cooled down to room temperature and depressurized. Next, hexane, water and a saturated solution of sodium hydrogen carbonate was added. The reaction mixture was extracted with EtOAc, dried over Na_2_SO_4_ and concentrated under reduced pressure. Purification by column chromatography using (Et_2_O:DCM:MeOH 2:7.5:0.5) as eluent gave **2** (38 mg, 82%) as a brown powder; mp 187.2–189.1 °C); Rf = 0.70 (Et_2_O:DCM:MeOH 2:7.5:0.5); ^1^H NMR (500 MHz, CHLOROFORM-d) δ = 7.93 (s, 1 H), 7.63–7.49 (m, 3 H), 7.15–7.03 (m, 3 H), 4.03 (s, 2 H), 2.96–2.92 (m, 3 H), 2.90 (s, 3 H), 2.36–2.28 (m, 3 H); ^13^C NMR (126 MHz, CHLOROFORM-d) δ = 168.2, 162.7 (d, *J* = 247.8 Hz), 143.7 (d, *J* = 23.8 Hz), 130.0, 129.8, 128.0 (d, *J* = 3.8 Hz), 122.2, 116.6, 115.6 (d, *J* = 3.0 Hz), 113.9, 37.6, 36.0, 30.1, 18.6; ^19^F NMR (471 MHz, CHLOROFORM-d) δ = −114.1 (s, 1 F); EI MS *m*/*z:* 312.0 [M + H^+^]. Alternatively, reaction performed in THF (3 mL) gave **2** with 75% yield.

*2-(6-fluoro-2-(4-fluorophenyl)imidazo[1,2-α]pyridin-1-yl)-N,N-dimethylacetamide (3).* To a solution of 2-6-fluoro-2-(4-fluorophenyl)imidazo[1,2-a]pyridin-3-yl)-2-hydroxy-*N*,*N*-dimethylacetamide **9** (50 mg, 0.151mmol, 1 eq) in dioxane (3 mL), phoshorus tribromide (0.039 mL, 0.393 mmol, 2.6 eq) was added. The mixture was irradiated at 90 °C for 45 min in the CEM-Discover system, operating at a frequency of 2.445 GHz, with continuous irradiation power from 0 to 200 W. The reactor was cooled down to room temperature and depressurized. Next, hexane, water and saturated solution of sodium hydrogen carbonate was added. The reaction mixture was extracted with EtOAc, dried over Na_2_SO_4_ and concentrated under reduced pressure. Purification by column chromatography using (Et_2_O:DCM:MeOH 2:7.5:0.5) as eluent gave **3** (36 mg, 76%) as a yellow powder; mp 183.2–185.4 °C; Rf = 0.70 (Et_2_O:DCM:MeOH 2:7.5:0.5); ^1^H NMR (500 MHz, CHLOROFORM-d) δ = 8.09–8.05 (m, 3 H), 7.63–7.59 (m, 4 H), 4.05 (s, 2 H), 2.97 (s, 3 H), 2.94 (s, 3 H), ^13^C NMR (126 MHz, CHLOROFORM-d) δ = 167.8, 163.1 (d, *J* = 248.6 Hz), 152.7 (d, *J* = 236.8 Hz), 146.2, 144.3, 130.4 (d, *J* = 7.3 Hz), 129.7 (d, *J* = 8.4 Hz), 118.2 (d, *J* = 9.7 Hz), 117.2 (d, *J* = 2.1 Hz), 115.7 (d, *J* = 16.2 Hz), 111.4, 111.3, 37.6, 35.9, 29.7; ^19^F NMR (235.3 MHz, CDCl3) δ: −112.7 (s, 1 F), −138.0; EI MS *m*/*z:* 316.0 [M + H^+^]. Alternatively, reaction performed in THF (3mL) gave 3 with 58% yield.

*N*,*N*-dimethyl-2-(6-methyl-2-(*p*-tolyl)imidazo[1,2-*a*]pyridin-3-yl)acetamide 2,3-dihydroxysuccinate (I) Method C.

To a solution of zolpidem base **1** (50 mg, 0.162 mmol, 2 eq) in dry methanol 2 ml, tartaric acid (12 mg, 0.081 mmol, 1 eq) was added, and the resulting mixture was stirred (520 rpm) for 2 h at reflux. After that time, the mixture was cooled to room temperature, next, 2 mL of diethyl ether was added in one portion. The addition of diethyl ether induced slow precipitation. The resulting mixture was placed in a freezer (at −14 °C) overnight, and the next day, 3 mL of diethyl ether was added, and the precipitate was washed with a small amount of diethyl ether (2 mL) to give **I** with 73% of yield (90 mg). The spectral data matched those previously reported [36]; white solid, mp 195–197 °C; ^1^H NMR (500 MHz, DMSO-d6) δ = 8.02 (d, *J* = 1.1 Hz, 1H), 7.55 (d, *J* = 8.4 Hz, 2H), 7.52 (d, J = 9.0 Hz, 1H), 7.25–7.21 (m, 2H), 7.08 (dd, *J* = 1.7, 9.2 Hz, 1 H), 4.30 (s, 1H), 4.12 (s, 2H), 3.10 (s, 3H), 2.81 (s, 3H), 2.32 (s, 3H), 2.28 (s, 3H); ^13^C NMR (126 MHz, CHLOROFORM-d) δ = 173.2, 169.1, 146.1,144.3, 137.7, 130.8, 128.2, 127.9, 122.4, 120.1, 117.6, 115.2, 71.8, 37.1, 35.4, 29.8, 21.1, 18.2; EI MS *m*/*z:* 308.0 [M + H^+^].

Sample preparation for microscope imagining: 1mg of zolpidem tartrate I was placed on a microscope plate, dissolved in a drop of methanol and evaporated directly on the microscope plate. The images were taken immediately after the solvent was evaporated using polarized light microscope Hund Wetzlar type H600/12 (Helmut Hund GmbH, Wetzlar, Germany).

## 4. Conclusions

We have disclosed a practical procedure for delivering lab-scale quantities of zolpidem **1** and its fluorinated **2**–**3** analogues, incorporating microware supported chemistry. The procedure allows for the effective preparation of the final products **1**–**3** in a minimum number of steps, and it starts from easily accessible and inexpensive starting materials, namely 2-aminopyridines and α-bromoacetophenones. Compared to conventional heating systems, our protocol shows superior performance in terms of yield and purity of products. Foremost, the application of microware chemistry substantially enhanced the speed of reaction performance and, thus, the overall time needed to complete the entire synthesis (from 36 h to 2.30 h). The disclosed method also utilizes a minimal amount of solvents in the reactions. Finally, we revealed a suitable method for preparing zolpidem tartrate salt **I**, and found that addition of a precipitant, ahead of crystallization, makes it possible to obtain a small-particle size of **I**, suitable for biological studies.

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
