# Peer review of "Study on a Three-Step Rapid Assembly of Zolpidem and Its Fluorinated Analogues Employing Microwave-Assisted Chemistry"

_molecules, 2020, doi:10.3390/molecules25143161_

Round 1

Reviewer 1 Report

The paper of Fajkis et.al. described the facile synthesis of Zolpidem and fluorinated derivatives using a microwave technique. The overall reaction scheme and explanation is well described and easy to read. The scope of the work is clear and the conclusion in line with the presentation in the introduction.

The analytical part of the work is complete and very clear all the compounds has been characterized by NMR and Mass spectrometry techniques and are in line with the molecules synthetized.

Only two small correction:

Page 1 line 28 GABA-eric must be substituted by GABA-ergic

Page 2 line 62 irritation must be substituted by irradiation.

In conclusion the paper is easy to read and really interesting. I suggest the publication after this minor revision.

Author Response

We would like to thank the Reviewer for his/her time and comprehensive review of our manuscript. With the aim of improving our work, we have tried to addressed all the comments and suggestions. We hope it will be able to meet with your approval.

According to the suggestion we have changed the typographic errors (page 1 and page 2)

Reviewer 2 Report

Marcinkowska and coworkers report an improved synthetic method of zolpidem for biological experimentation in this manuscript. The synthetic methods of biological probes are essential. Authors developed the synthetic route of zolpidem in their previous study (Eur, J. Med. Chem. 2026, 124, 456.). Herein, they improved the protocol using Microwave technique. Reaction time as well as the yields were improved. These findings are useful for readers. The findings for the precipitation of the final product is also important. This reviewer supports the acceptation of this manuscript.

Additional comments,

In table 1, the presented time unit is "h". Is this correct?

In table 6, the results of Method B and C were opposite?  The results in the text are not consistent with that in the table.

This reviewer requests that details of precipitation conditions (Method C) are added in experimental section for the reproducibility: the rate of adding Et2O (in one portion?), stirring rate, the temperature of refrigerator.

I don't think "green" in conclusions is suitable. DCM was used for purification. 

There are several typos. In page 6, PPBr3, dihydroxylation, in page 13 dietyleter.

Author Response

We would like to thank the Reviewer for his/her time and comprehensive review of our manuscript. With the aim of improving our work, we have tried to addressed all the comments and suggestions. We hope it will be able to meet with your approval.

Marcinkowska and coworkers report an improved synthetic method of zolpidem for biological experimentation in this manuscript. The synthetic methods of biological probes are essential. Authors developed the synthetic route of zolpidem in their previous study (Eur, J. Med. Chem. 2026, 124, 456.). Herein, they improved the protocol using Microwave technique. Reaction time as well as the yields were improved. These findings are useful for readers. The findings for the precipitation of the final product is also important. This reviewer supports the acceptation of this manuscript.

Additional comments,

In table 1, the presented time unit is "h". Is this correct?

We have placed accidentally “h” in the table 1. The error has been corrected.

In table 6, the results of Method B and C were opposite?  The results in the text are not consistent with that in the table.

We have accidentally replaced the description “yield” with “purity” in the table 6. The error has been corrected.

This reviewer requests that details of precipitation conditions (Method C) are added in experimental section for the reproducibility: the rate of adding Et2O (in one portion?), stirring rate, the temperature of refrigerator.

According to the suggestion we have provided detailed description of the Method C, which can be found in the experimental section (Page 12).

I don't think "green" in conclusions is suitable. DCM was used for purification. 

According to the suggestion we have modified the description in the conclusions.

There are several typos. In page 6, PPBr3, dihydroxylation, in page 13 dietyleter.

According to the suggestion we have changed all the typographic errors.